# Navigating the Effects of Anti-Atherosclerotic Supplements and Acknowledging Associated Bleeding Risks

**DOI:** 10.3390/ijms262010183

**Published:** 2025-10-20

**Authors:** Maria-Zinaida Dobre, Bogdana Virgolici, Ioana-Cristina Doicin, Horia Vîrgolici, Iulia-Ioana Stanescu-Spinu

**Affiliations:** 1Department of Biochemistry, Faculty of Medicine, Carol Davila University of Medicine and Pharmacy, 050474 Bucharest, Romania; maria.dobre@umfcd.ro; 2Cardiology Resident at the Central Military Emergency University Hospital “Dr. Carol Davila”, 88 Mircea Vulcanescu Street, 010825 Bucharest, Romania; ioana-cristina.doicin@rez.umfcd.ro; 3Department of Marketing and Medical Technology, Faculty of Medicine, Carol Davila University of Medicine and Pharmacy, 050474 Bucharest, Romania; horia.virgolici@umfcd.ro; 4Department of Physiology, Faculty of Dentistry, Carol Davila University of Medicine and Pharmacy, 050474 Bucharest, Romania; iulia.stanescu@umfcd.ro

**Keywords:** nutraceuticals, cardiovascular health, omega-3 fatty acids, berberine, garlic, nattokinase, antioxidants, bleeding risk

## Abstract

Several nutraceuticals demonstrate potential cardiovascular benefits through lipid-lowering, antithrombotic, and vascular protective mechanisms. Omega-3 fatty acids, berberine, garlic, and nattokinase exert favorable metabolic and vascular effects, yet their clinical efficacy depends on formulation, dosage, and patient characteristics and may be limited by bleeding risk or drug interactions. Antioxidant agents such as vitamin C, vitamin E, resveratrol, astaxanthin, and coenzyme Q provide additional vascular protection but can interfere with hemostasis, metabolism, or redox-sensitive pathways. Similarly, ginkgo biloba, ginger, ginseng, and curcumin exhibit anti-inflammatory vascular activity but also increase the risk of bleeding when combined with antithrombotic therapy. Given the variability in evidence and product quality, their use should be individualized, with further large-scale clinical trials needed to establish safety and efficacy.

## 1. Introduction

The development of atherosclerosis, an essential promoter of cardiovascular diseases and stroke via thrombus formation [1], is supported by the interplay between platelets and endothelial cells [2]. Taking into account that atherosclerosis is considered to be the main cause of mortality on a global scale, its pathogenesis has been a hot research topic for decades [3]. In order to reduce the risk of thrombosis, patients with atherosclerotic cardiovascular disease are often prescribed antiplatelet therapy. Nevertheless, platelet aggregation inhibition can also result in an increased bleeding risk, which is why the clinical approach must consider the benefit of single or dual antiplatelet treatments. The bleeding risk in atherosclerosis can arise from plaque rupture or the presence of intraplaque hemorrhage.

Vascular chronic inflammation, which leads to the formation of atheroma plaque and its rupture, begins in the endothelium of the blood vessels, where cells disposed on a single layer, called endothelial cells (EC), detect signals produced by biologically active substances and transmit them to the vascular wall [4].

The main events that characterize the development of atherosclerosis are as follows: endothelial cells are activated as a result of damage to the glycocalyx, which loses its function as a barrier and increases its permeability for low-density lipoprotein cholesterol (LDC-C); leukocytes, especially monocytes, which are transformed into macrophages, internalize oxidized LDL-C (ox-LDL-C) and become foam cells; and atheroma plaque and the atheromatous nucleus are formed [5]. In atherosclerosis, leukocyte activation induced by platelets leads to vascular inflammation, elevated endothelial permeability and plasma protein accumulation in the interstitium, events promoted by upregulation of intercellular adhesion molecules (ICAM), vascular cell adhesion molecule 1 (VCAM-1) and E-selectin [6]. The resulting EC damage leads to an imbalance between vasodilation and vasoconstriction processes, reactive oxygen species (ROS) formation and cytokine release (IL-1, IL-6, TNF-alpha), decreased NO availability and finally endothelial dysfunction and thrombosis [7,8]. Upon plaque rupture, platelet aggregation is stimulated, and the thrombus can result in ischemia and infarction [9].

P-selectin together with von-Willbrand factor are responsible for the interaction of platelets with activated endothelial cells, leading to platelet activation. Moreover, P-selectin aggravates atherosclerosis by distributing the proinflammatory mediators produced by platelets to the vascular wall [10]. Binding P-selectin present on platelets to an analogue receptor called P-selectin glycoprotein ligand-1 (PSGL-1) results in the formation of leukocyte–platelet aggregates. As a result, leukocyte β2 integrins macrophage antigen-1 (Mac-1) and lymphocyte function-associated antigen-1 (LFA-1) are activated and platelets produce chemokines, including platelet factor 4 (PF4), a vital mediator of atherosclerosis, to enhance the adhesion of monocytes, which will further turn into macrophages and will finally become foam cells by lipid uptake [11]. PF4 directly favors the internalization of ox-LDL-C by macrophages and prevents the destruction of the receptor for LDL-C [12].

Therefore, in atherosclerosis, a significant role is played by platelets, also known as thrombocytes, which interact with ECs, progenitor cells, and leukocytes, playing a significant pro-inflammatory role [13]. In fact, in atherosclerosis, they are considered the connection between plaque formation, thrombosis and inflammation, as well as intervening after plaque rupture [14]; a meta-analysis of 50 randomized clinical trials showed that antiplatelet and anticoagulant therapies increase the bleeding risk [15]. Therefore, platelets are not only responsible for promoting the development of atherosclerotic disease but could also play key roles as therapeutic targets in order to reduce the risk of complications, including decreasing the bleeding risk [16].

Furthermore, in atherosclerosis, neovascularization (vasa vasorum) in the arterial wall is favored by the release of vascular endothelial growth factor (VEGF), cytokines and chemokines from platelets. Plaque immature new blood vessels present abnormal structural and functional properties, with discontinuation in the base membrane and decreased normal junctions between the ECs, and promote plaque instability and intraplaque hemorrhage, usually in the proximity of the necrotic core [17].

However, hemorrhage can also occur, favored by inflammation, which causes endothelial damage of mature blood vessels, mediated by free hemoglobin, which stimulates ROS formation [18]. Previous studies revealed a link between lesion instability and intraplaque hemorrhage [19], which can lead to rupture and cause internal bleeding.

Figure 1 summarizes the atherothrombotic cascade, highlighting how supplements exert anti-atherosclerotic and antiplatelet effects that may overlap and influence bleeding risk.

## 2. Nutraceuticals and Cardiovascular Health: Anti-Atherosclerotic Potential and Safety Concerns

The use of dietary supplements for cardiovascular disease (CVD) prevention has risen in recent years, often promoted as natural alternatives or adjuncts to lipid-lowering therapies. Nutraceuticals—bioactive compounds from foods or plants—are of particular interest. Agents such as omega-3 fatty acids, plant sterols, soluble fibers, niacin, red yeast rice, and berberine have demonstrated lipid-lowering effects, including reductions in total cholesterol, LDL-C, and triglycerides [20].

However, clinical benefits remain uncertain. Evidence from randomized trials and meta-analyses shows wide variability, influenced by supplement composition, dosage, study design, and patient characteristics [21]. In the United States, supplements are regulated under the FDA’s Dietary Supplement Health and Education Act (DSHEA), which imposes limited requirements for safety, efficacy, and labeling, raising public health concerns [22,23]. Despite this, use remains widespread, often driven by low consumer awareness.

In selected patients, such as those with statin intolerance or seeking integrative approaches, certain supplements may provide incremental benefit, though routine use for CVD prevention is not supported by high-quality evidence [24,25]. Importantly, nutraceuticals should not replace proven therapies.

Some supplements promoted for anti-atherosclerotic effects also possess antiplatelet activity and may increase bleeding risk, particularly in patients receiving anticoagulant or antiplatelet drugs. Examples include omega-3 fatty acids, garlic, ginkgo biloba, ginseng, curcumin, vitamin E, ginger, green tea extract, and high-dose fish oil. Caution is advised in patients on anticoagulation or undergoing surgery.

This review highlights mechanisms by which dietary supplements may exert anti-atherosclerotic effects and identifies those associated with elevated bleeding risk, especially in patients already receiving antithrombotic therapy.

### 2.1. Anti-Atherosclerotic Supplements Acting as Lipid Lowering Agents

#### 2.1.1. Omega-3 Fatty Acids

Among the most extensively studied nutraceuticals for lipid modulation are omega-3 fatty acids, particularly eicosapentaenoic acid (EPA) and docosahexaenoic acid (DHA). These long-chain polyunsaturated fatty acids, obtained primarily from marine sources such as fish and krill oil, have consistently demonstrated the ability to reduce serum triglyceride levels by 20–50%, depending on dosage and baseline lipid status [26]. The proposed mechanisms of action include inhibition of hepatic very-low-density lipoprotein (VLDL) synthesis, enhancement of fatty acid β-oxidation, and suppression of hepatic lipogenesis [27,28].

Beyond their lipid-modifying properties, omega-3 fatty acids exert antiplatelet effects primarily through inhibition of thromboxane A_2_ (TxA_2_) synthesis and attenuation of TxA_2_-dependent platelet activation. This occurs through competition between arachidonic acid and EPA for enzymatic pathways involved in thromboxane biosynthesis, as well as through the generation of EPA-derived metabolites, such as resolvins, which interact with TxA_2_ receptors and modulate their activity [29,30,31,32].

EPA and DHA also appear to influence platelet function by reducing membrane fluidity, which decreases the externalization of procoagulant phospholipids, particularly phosphatidylserine. This alteration leads to impaired calcium influx, thereby attenuating activation of the coagulation cascade and inhibiting platelet aggregation [33,34].

In addition, omega-3 fatty acids are presumed to modulate platelet activation through effects on the glycoprotein VI (GPVI) collagen receptor. Proposed mechanisms include direct interference with GPVI signaling pathways or attenuation of collagen-dependent platelet reactivity via protein kinase A activation [35,36,37].

Atherosclerosis is closely linked to the adiponectin/leptin ratio, an important marker of metabolic and inflammatory balance. In a systematic review, nine studies reported significantly lower leptin and/or higher adiponectin levels following EPA + DHA intake. The doses, supplementation duration, and population characteristics varied across studies. The EPA + DHA dose ranged from 0.52 to 4.2 g/day, with supplementation lasting 4 to 24 weeks [38].

Table 1 highlights key clinical studies from the most recent trials addressing the potential benefits of omega-3 fatty acids in atherosclerosis, including triglyceride reduction, anti-inflammatory effects, improved endothelial function, and plaque stabilization.

#### 2.1.2. Berberine

Berberine, an isoquinoline alkaloid from Coptis chinensis and Berberis species, exhibits broad pharmacological activity with therapeutic potential in cancer, metabolic, cardiovascular, digestive, and neurological diseases [44]. Berberine improves glucose and lipid metabolism by activating AMPK in adipocytes and myocytes, which enhances GLUT4 translocation, boosts lipid oxidation, and inhibits lipid synthesis [45]. Berberine has a lipid-lowering effect by stabilizing LDLR mRNA, thereby increasing LDL receptor levels and enhancing LDL cholesterol clearance from the blood [46]. Additionally, berberine reduces cholesterol levels by inhibiting its intestinal absorption and limiting its release into the bloodstream [47].

In vitro, berberine and its metabolite berberrubine (M2) inhibited ADP-induced platelet activation by blocking Glycoprotein IIb/IIIa activation, reducing P-selectin expression, and limiting fibrinogen binding. Both compounds selectively targeted PI3Kβ, likely through interaction with its active site [48].

Network pharmacology analysis revealed that berberine acts against atherosclerosis through pathways involving the cell cycle, ubiquitin-mediated proteolysis, MAPK, and PI3K-Akt signaling. Overall, these mechanisms indicate that berberine may reduce atherosclerosis by inhibiting inflammation and vascular cell proliferation [49].

#### 2.1.3. Red Yeast Rice

Red yeast rice (RYR) is a fermented product obtained by growing the fungus Monascus purpureus on rice, producing bioactive compounds such as monacolin K and pigments [50]. Similar to traditional statins, monacolins inhibit HMG-CoA reductase resulting in reduction of endogenous cholesterol synthesis and reduced blood cholesterol levels [51]. RYR can lower LDL-C to a degree comparable to low-dose statins and is particularly useful for individuals with mild-to-moderate hypercholesterolemia who are not eligible for statins, as well as for those who are intolerant to statins or prefer nutraceuticals [52]. RYR may also help avoid some of the ‘nocebo’ effects observed with statins [53].

Liu et al. demonstrated that combined nattokinase and red yeast rice supplementation safely enhances cardiometabolic profiles in patients with coronary artery disease, showing stronger reductions in thromboxane B2 and increases in antithrombin III than placebo, suggesting lowered thrombosis risk [54]. Moreover, this year, a double-blind, parallel-controlled trial is being conducted to evaluate the effect of a Natto Red Yeast Rice (NRYR) supplement combined with statins on lipid levels in individuals with dyslipidemia [55].

#### 2.1.4. Garlic

Garlic (*Allium sativum*) has a long-standing history of medicinal use, including for cardiovascular protection. Its active compounds—particularly allicin and S-allyl cysteine—have lipid-lowering, anti-inflammatory, and antioxidative effects [56].

Garlic’s lipid-lowering effects are multifactorial, involving inhibition of hepatic cholesterol synthesis—potentially through suppression of HMG-CoA reductase—enhancement of biliary excretion of cholesterol and bile acids, and antioxidant activity that reduces LDL oxidation and atherosclerotic plaque formation [57].

An early systematic review reported an average 12% reduction in total cholesterol, supporting garlic’s modest but consistent lipid-lowering effect after more than four weeks of supplementation [58]. A meta-analysis of 14 studies (1981–2016) further confirmed significant decreases in total cholesterol and LDL-C, though effects on HDL and triglycerides were not significant [57]. A recent 2024 meta-analysis by Du et al., including 21 randomized controlled trials, found that garlic significantly reduced total cholesterol, LDL-C, and triglycerides, with a slight increase in HDL-C [59]. The observed heterogeneity in study outcomes was likely influenced by differences in garlic formulation, dosage, treatment duration, and participant characteristics, as well as methodological limitations such as the absence of subgroup analyses [57,59]. Thus, greater lipid-lowering effects were observed in individuals with high baseline cholesterol, particularly with aged garlic extract, which also improved oxidative stress markers [60].

Garlic also modulates gut microbiota composition, which may indirectly influence lipid metabolism and bile acid homeostasis [61].

Garlic extract and other phytochemicals may regulate lipid homeostasis by modulating the mevalonate pathway, which is key to cholesterol metabolism and tumorigenesis, offering potential benefits for dyslipidemia, obesity-related complications, and cancer [62].

Allicin improves metabolic function in diabetic rats by lowering blood glucose and lipids, reducing liver fat, modulating gut microbiota and bile acids, increasing intestinal GLP-1, suppressing FGF15, and upregulating hepatic CYP7A1 [61].

Garlic inhibits platelet aggregation by suppressing cyclooxygenase activity and thromboxane A_2_ synthesis, reducing intraplatelet Ca^2+^, increasing cAMP and cGMP, and decreasing fibrinogen binding and platelet shape change [63].

In addition, its antioxidant properties and stimulation of nitric oxide synthase (NOS) enhance platelet-derived NO production. Garlic compounds may also directly interfere with GPIIb/IIIa receptors, reducing platelet–fibrinogen interactions [60]. Garlic exerts anti-inflammatory effects by modulating cytokine production and improving endothelial function [40].

#### 2.1.5. Nattokinase

Natto is a traditional Japanese food produced by fermenting soybeans with Bacillus natto. This fermentation generates several bioactive compounds, among which nattokinase is the most studied. Nattokinase (NK) is a serine protease known for its enzymatic stability and resistance to freeze–thaw cycles. It contributes to lipid regulation and anti-atherosclerotic activity by stimulating hormone-sensitive lipase, suppressing HMG-CoA reductase, and promoting lipoprotein lipase function [64].

By combining anti-inflammatory, antioxidant, hemorheological, and thrombolytic actions with plaque stabilization and endothelial protection, nattokinase emerges as a promising cardioprotective agent. Through modulation of TLR-4/JAK-STAT and TRAF-6/NF-κB-MAPK signaling, nattokinase decreases IL-6 mediated inflammation and ROS-driven caspase activation, ultimately reducing apoptosis in vascular and myocardial tissue [65]. Nattokinase supplementation has no impact on the progression of subclinical atherosclerosis in healthy, low–cardiovascular-risk individuals [66].

Nattokinase not only degrades fibrin directly but also enhances fibrinolysis by increasing tissue plasminogen activator (tPA) release and inactivating PAI-1 [67].

It facilitates the conversion of prourokinase to urokinase, further promoting clot lysis [68]. In addition, NK suppresses thromboxane formation, thereby inhibiting platelet aggregation without inducing bleeding [69]. Human studies have shown reductions in fibrinogen, factor VII, and factor VIII, as well as shortened euglobulin lysis time and increased fibrin/fibrinogen degradation products, indicating potent anticoagulant and fibrinolytic effects [70].

In a clinical trial involving 113 patients with dyslipidemia, participants were randomly assigned to receive either nattokinase-monascus supplementation or placebo. Treatment with nattokinase-monascus supplementation significantly reduced total cholesterol and low-density lipoprotein cholesterol, while no significant effects on coagulation parameters were observed [71].

### 2.2. Nutraceutical and Botanical Bioactives That Inhibit LDL Oxidation

The oxidative modification of LDL into its atherogenic form is widely recognized as a critical event in the initiation and progression of atherosclerosis [8]. Oxidized LDL (oxLDL) disrupts endothelial homeostasis, intensifies pro-inflammatory signaling, and accelerates foam cell formation within the arterial wall [72]. A range of bioactive molecules—including plant-derived antioxidants and lipid-phase cofactors such as coenzyme Q10—have been investigated for their ability to counteract this process by directly scavenging reactive species or by strengthening endogenous redox defenses. When optimized in terms of formulation and bioavailability, these agents may offer complementary vascular protection, though potential pharmacokinetic interactions with conventional cardiovascular therapies must be considered [73].

#### 2.2.1. Astaxanthin

Astaxanthin is a lipid-soluble xanthophyll carotenoid with a high affinity for cellular membranes, where it integrates into lipoproteins and phospholipid bilayers to protect polyunsaturated fatty acids from oxidative degradation. This spatial localization enables efficient quenching of singlet oxygen and lipid radicals, thereby terminating the propagation phase of lipid peroxidation [74]. Clinical investigations—particularly in individuals with impaired glucose metabolism and elevated oxidative stress—have reported enhanced LDL oxidative stability and reductions in lipid peroxidation markers such as malondialdehyde following supplementation with astaxanthin at 12 mg/day for 12 weeks [75,76,77]. Although no clinically significant drug–nutrient interactions have been established, the strong redox activity of astaxanthin suggests a theoretical potential to influence the pharmacokinetics of redox-sensitive agents, a hypothesis warranting further mechanistic investigation [78]. Small randomized studies indicate improvements in oxidative stress markers and LDL stability; however, multicenter outcome trials confirming cardiovascular benefits are still lacking [79,80,81].

#### 2.2.2. Resveratrol

Resveratrol, a stilbenoid polyphenol found abundantly in red grapes and Polygonum cuspidatum, supports vascular health by reducing oxidative stress and enhancing endogenous antioxidant defenses [82]. It inhibits NADPH oxidase and stimulates antioxidant enzymes such as superoxide dismutase and catalase, thereby mitigating the oxidative modifications that render LDL particles atherogenic [83,84]. In addition, resveratrol modulates lipid homeostasis by activating the EGFR–ERK signaling pathway, leading to upregulation of hepatic LDL receptor expression and improved clearance of circulating LDL, which may slow the development of early atheromatous plaques [85]. Preclinical studies suggest potential synergy with statins through enhanced endothelial repair and accelerated vascular healing following injury [86]. Mechanistic investigations further demonstrate that resveratrol restores autophagic flux in endothelial cells exposed to oxidized LDL by upregulating SIRT1, promoting lysosomal degradation pathways, and reducing oxLDL accumulation, underscoring a reparative role that extends beyond its classical antioxidant activity [87].

Small randomized trials and meta-analyses in humans report reductions in inflammatory biomarkers (e.g., CRP, TNF-α) and improvements in endothelium-dependent vasodilation assessed by flow-mediated dilation (FMD), albeit with heterogeneous effects across studies; however, large multicenter outcome trials showing reductions in major adverse cardiovascular events are lacking [88,89,90,91].

#### 2.2.3. Coenzyme Q10 (CoQ10)

Coenzyme Q10 (Ubiquinone/Ubiquinol) functions both as an essential cofactor in the mitochondrial electron transport chain and as a lipid-phase antioxidant that protects cellular and lipoprotein membranes. By integrating into LDL particles, CoQ10 helps preserve structural integrity under oxidative stress, thereby limiting the formation of atherogenic oxidized LDL (oxLDL). Clinically, supplementation at doses of 200–400 mg/day has been associated with reductions in circulating oxLDL and improvements in endothelium-dependent vasodilation, as assessed by flow-mediated dilation (FMD), particularly among statin-treated patients [92]. This is mechanistically relevant because statins inhibit the mevalonate pathway, reducing endogenous CoQ10 synthesis and potentially aggravating mitochondrial dysfunction [93]. Restoring CoQ10 levels is therefore a rational adjunctive strategy to mitigate statin-associated oxidative and bioenergetic deficits. Recent meta-analyses support these findings, reporting significant improvements in FMD with CoQ10 supplementation, consistent with enhanced endothelial function [94,95]. In a randomized controlled trial among statin users, CoQ10 improved mitochondrial function and antioxidant capacity, although effects on muscle symptoms were variable [96]. Overall, CoQ10 exhibits a favorable safety profile when used at customary doses.

Meta-analyses indicate modest improvements in endothelium-dependent vasodilation assessed by flow-mediated dilation (FMD) and oxidative biomarkers—particularly in statin-treated populations—yet robust atherosclerotic outcome data are limited. A multicenter RCT in chronic heart failure (Q-SYMBIO) reported reductions in major adverse cardiovascular events with adjunctive CoQ10, but its population and endpoints differ from atherosclerosis prevention [94,96,97,98,99]. In elderly with low selenium status, combined selenium + CoQ10 reduced cardiovascular mortality in the randomized KiSel-10 trial and its long-term follow-ups; however, effects cannot be attributed to CoQ10 alone [100,101,102]. A recent RCT in MASLD reported improvements in vascular and myocardial function with high-dose CoQ10, extending evidence to a metabolic-risk cohort, though without hard CV endpoints [103].

#### 2.2.4. Vitamins C and E

Vitamins C and E act synergistically to limit oxidative vascular injury through complementary mechanisms. Vitamin E refers to a family of lipid-soluble tocopherols and tocotrienols that incorporate into lipoproteins and cellular membranes, where α-tocopherol terminates lipid peroxidation chain reactions and reduces LDL susceptibility to oxidative modification [104]. Tocotrienols may confer additional benefits: in a 2025 randomized trial, a tocotrienol-rich formulation (300 mg/day) significantly lowered LDL-C and C-reactive protein compared with α-tocopherol–dominant preparations [105]. The inter-vitamin recycling mechanism—where ascorbate regenerates oxidized α-tocopherol—underlies the observed synergy during co-supplementation, yielding up to ~40% greater reductions in lipid peroxidation compared with either vitamin alone [106]. Clinical studies employing approximately 1 g/day of vitamin C have reported reductions in inflammatory biomarkers (CRP, IL-6), decreases in lipid peroxidation markers (e.g., malondialdehyde), and improvements in flow-mediated dilation (FMD) among patients with diabetes or metabolic syndrome [107].

Across randomized trials and meta-analyses, vitamins C and E yield heterogeneous effects on inflammatory and oxidative stress biomarkers and modest improvements in endothelium-dependent vasodilation assessed by flow-mediated dilation (FMD); however, large multicenter trials have not consistently demonstrated reductions in major cardiovascular events with high-dose vitamin C or E [108,109,110,111]. High-dose vitamin E (≥400 IU/day) has also been associated with increased all-cause mortality in a dose–response meta-analysis [112,113].

### 2.3. Supplements with Anti-Atherosclerotic Effects That Reduce Vascular Inflammation

#### 2.3.1. Ginkgo Biloba

Ginkgo biloba (*Maidenhair tree*) is an ancient, long-lived species valued for its distinctive fan-shaped leaves and medicinal properties. EGb 761, a standardized extract of Ginkgo biloba leaves, is widely used in phytotherapy and clinical research. Its principal constituents—ginkgolide B and flavonoids—upregulate endothelial nitric oxide synthase (eNOS) expression and promote Akt-dependent eNOS phosphorylation, thereby enhancing nitric oxide (NO) release and vasorelaxation [114]. At the molecular level, ginkgolide B mitigates endothelial oxidative stress by downregulating LOX-1 (lectin-like oxidized LDL receptor 1) and NOX4 (NADPH oxidase 4), leading to reduced reactive oxygen species (ROS) generation and decreased expression of vascular adhesion molecules (ICAM-1, VCAM-1, MCP-1) [115]. In vitro, Ginkgo biloba extract suppresses cytokine-induced endothelial adhesiveness by upregulating heme oxygenase-1 (HO-1) through activation of the p38 and Nrf2 signaling pathways. In vivo, it reduces leukocyte adhesion to injured arteries and increases HO-1 expression in monocytes and arterial tissue following wire-induced injury [116]. Furthermore, EGb 761 activates HO-1 in endothelial progenitor cells, supporting re-endothelialization and vascular repair [117].

Human evidence derives mainly from small, short-duration randomized trials with standardized EGb 761 (typically 120–240 mg/day over 4–24 weeks). These studies show modest, inconsistent improvements in endothelium-dependent vasodilation assessed by FMD and microvascular perfusion surrogates, with neutral or variable effects on lipids, inflammation, or blood pressure; trials are heterogeneous in extract, dose, and endpoints and are underpowered for clinical events. Crucially, no large multicenter RCT has demonstrated reductions in myocardial infarction, stroke, or cardiovascular death: in the 3069-participant GEM program (EGb 761, 120 mg twice daily), secondary analyses found no reduction in incident CVD or CVD mortality [118,119,120,121]. Recent clinical work continues to focus on surrogate and inflammatory/oxidative markers rather than hard cardiovascular outcomes (e.g., randomized/open-label EGb 761 in mild cognitive impairment reporting changes in circulating inflammation/oxidative stress biomarkers). These newer trials reinforce the surrogate-heavy and underpowered nature of the literature [122,123].

#### 2.3.2. Ginger

Ginger (*Zingiber officinale*) is a widely used medicinal and culinary plant, valued for its bioactive phenolic constituents—6-gingerol and 6-shogaol—which exhibit potent anti-inflammatory and antioxidant properties. These compounds attenuate systemic inflammation by lowering circulating levels of C-reactive protein (CRP), interleukin-6 (IL-6), and tumor necrosis factor-α (TNF-α), biomarkers consistently associated with endothelial dysfunction [124]. At the cellular level, 6-gingerol suppresses reactive oxygen species (ROS) production and restores redox homeostasis in endothelial cells exposed to oxidative stress, in part through activation of the Nrf2/HO-1 signaling pathway [125].

Meta-analyses of randomized trials report that ginger supplementation (typically 1–3 g/day for 8–12 weeks) significantly reduces hs-CRP and TNF-α across diverse adult populations, with inconsistent effects on IL-6 [126]. In metabolic-risk cohorts (e.g., metabolic syndrome, type 2 diabetes), ginger has been associated with lower triglycerides, improved fasting glycemia/HbA1c, and enhanced systemic antioxidant status, while effects on blood pressure or HDL/LDL cholesterol are variable [127]. Collectively, these changes are compatible with improved endothelial milieu and vasodilator responsiveness. The clinical literature is dominated by small RCTs and meta-analyses of surrogate outcomes (inflammation, oxidative stress, glycemic indices, lipids). While signals are directionally favorable for hs-CRP/TNF-α and selected metabolic parameters, large multicenter outcome trials showing reductions in myocardial infarction, stroke, or cardiovascular death are not yet available [126,128,129]. Most RCTs used powdered/extract ginger at 1–3 g/day for 8–12 weeks [129].

#### 2.3.3. Ginseng

Asian ginseng *(Panax ginseng*) is a medicinal plant traditionally used in East Asia, whose principal saponins—ginsenosides (e.g., Rg3, Rb1, Rd, Rp1, Rp3)—exert adaptogenic and immunomodulatory effects [130,131]. Ginsenosides activate PI3K/Akt–AMPK–eNOS signaling, enhance nitric oxide (NO) production, and suppress endothelin-1 expression, thereby promoting vasorelaxation. Antioxidant defenses are strengthened via Nrf2/HO-1 induction, while NF-κB signaling and endothelial adhesion molecules (ICAM-1, VCAM-1) are downregulated in experimental models [132]. Ginseng also modulates platelet-related pathways (e.g., GPVI–PLCγ2, PI3K/Akt, MAPKs), which may contribute to reduced thrombo-inflammatory activation and improved plaque stability [133]. Small randomized trials have reported modest improvements in endothelium-dependent vasodilation, as assessed by flow-mediated dilation (FMD) and other vascular surrogates, with heterogeneous effects on blood pressure and lipid profiles. Examples include acute FMD enhancement after administration of 3 g Korean red ginseng in healthy adults and variable findings among individuals with prehypertension or metabolic risk [134,135,136]. Clinical literature largely comprises small, short-duration randomized controlled trials (RCTs) employing surrogate endpoints; results vary according to preparation and dose, and most studies are underpowered for major cardiovascular outcomes. Meta-analyses focusing on vascular surrogates suggest improvements in FMD; however, no large multicenter RCTs have demonstrated reductions in myocardial infarction, stroke, or cardiovascular mortality [133,137,138].

#### 2.3.4. Curcumin

Curcumin/turmeric (*Curcuma longa*) is a medicinal and culinary plant whose rhizome contains curcumin, a polyphenol with potent anti-inflammatory and antioxidant properties. At the endothelial level, curcumin suppresses NF-κB activity, thereby reducing the expression of adhesion molecules (ICAM-1, VCAM-1) and limiting cytokine release (IL-6, TNF-α) [139]. Concurrently, it activates Nrf2-dependent antioxidant pathways, enhancing heme oxygenase-1 (HO-1) and other cytoprotective enzymes to restore redox balance under oxidative stress [140]. Clinical findings broadly support these mechanisms. An umbrella meta-analysis of randomized trials (2024) reported modest but significant vascular improvements with curcumin supplementation, including reductions in diastolic blood pressure (−0.94 mmHg), pulse wave velocity, and circulating VCAM-1, along with an approximate 1.6% increase in flow-mediated dilation (FMD) [141]. Benefits were more consistent in studies employing optimized formulations (e.g., nanoparticles, phytosomes, phospholipid complexes) designed to overcome curcumin’s poor oral bioavailability [142,143].

Across small-to-moderate randomized controlled trials (RCTs) and meta-analyses, curcumin demonstrates modest improvements in endothelial surrogates—most consistently in endothelium-dependent vasodilation assessed by FMD, arterial stiffness (PWV), and reductions in VCAM-1—accompanied by favorable changes in inflammatory and oxidative biomarkers. However, large multicenter outcome trials confirming reductions in myocardial infarction, stroke, or cardiovascular mortality remain unavailable [141,144,145,146].

## 3. Hemostatic Safety of Nutraceutical and Botanical Bioactives: Bleeding Risks and Drug–Drug Interactions

Nutraceutical and botanical use is common yet often under-disclosed, with perioperative societies advising discontinuation 1–2 weeks before elective procedures; formulation variability and limited standardization further complicate risk appraisal [147]. This section synthesizes hemostatic safety signals across three domains: (1) pharmacodynamic antiplatelet and fibrinolytic effects (e.g., Ginkgo biloba, ginger, curcumin, resveratrol, omega-3 fatty acids, garlic); (2) micronutrient-related effects on coagulation, particularly with high-dose vitamin E; and (3) pharmacokinetic interactions influencing warfarin or direct oral anticoagulant (DOAC) exposure [148].

### 3.1. Pharmacodynamic (Antiplatelet/Fibrinolytic) Effects

**Ginkgo biloba** standardized extracts (EGb 761) contain terpene trilactones—most notably ginkgolide B—which act as potent antagonists of the platelet-activating factor (PAF) receptor, a well-established antiplatelet mechanism demonstrated in biochemical and platelet models [149,150]. Authoritative safety sources therefore caution against concurrent use of ginkgo with aspirin, clopidogrel, nonsteroidal anti-inflammatory drugs (NSAIDs), or warfarin due to the potential for additive antiplatelet effects and altered coagulation parameters [151]. In large clinical datasets, concomitant use with vitamin K antagonists has been associated with an increased risk of major bleeding—for example, a chart review of approximately 807,399 patients found higher rates of major bleeding with ginkgo plus warfarin compared with warfarin alone [152], while a Veterans Administration electronic health record cohort reported a 38% higher hazard of bleeding with co-prescribed ginkgo (HR 1.38, 95% CI 1.20–1.58) [153]. Beyond vitamin K antagonists, a 2025 hospital-based retrospective analysis identified that ginkgo interactions most frequently involved aspirin or clopidogrel and were associated with abnormal coagulation tests and a small but significant increase in clinical bleeding events [154]. Complementing these observations, case reports have described intracranial and ocular hemorrhages temporally associated with ginkgo use, typically in patients receiving concurrent antithrombotic therapy [155].

**Ginger’s** principal phenolics (6-gingerol and 6-shogaol) inhibit platelet thromboxane synthesis and attenuate cyclooxygenase-1 (COX-1)-dependent aggregation, with human and ex vivo evidence showing reduced platelet reactivity [156,157]. Clinically, concurrent warfarin and ginger use has been linked to altered anticoagulation and bleeding in case reports, including supratherapeutic INR and prolonged bleeding, suggesting a plausible pharmacodynamic interaction [158]. In a prospective cohort of warfarin users, ginger intake was independently associated with a higher risk of self-reported bleeding [159]. However, expert reviews note that at modest doses, ginger (and ginkgo) does not consistently affect warfarin activity, highlighting dose, preparation, and patient context as key modifiers and supporting a cautious, individualized approach [160]. In clinical practice, drug-interaction compendia and perioperative guidelines advise avoiding concomitant use with antiplatelet or anticoagulant agents when feasible and discontinuing ginger 1–2 weeks before elective procedures, given the potential for additive bleeding risk [161].

**Curcumin’s** principal actions on platelets include suppression of thromboxane-dependent aggregation and inhibition of downstream signaling (Akt/MAPK/Src) with blockade of dense-granule secretion—mechanisms demonstrated in human/ex vivo models [162,163]. Clinically, the bleeding signal emerges largely in the setting of concomitant antithrombotic therapy, with pharmacovigilance alerts and case reports describing supratherapeutic INR and hemorrhagic events when turmeric/curcumin is used alongside warfarin or fluindione [164,165]. Randomized data quantifying bleeding outcomes are lacking—the evidence base is predominantly mechanistic, case-based, or observational—which underpins conservative perioperative and cardiovascular guidance [166]. In practice, co-administration with antiplatelet agents or warfarin warrants caution, and theoretical DOAC interactions remain a concern given curcumin’s inhibition of P-glycoprotein and reported modulation of CYP3A4, pathways central to apixaban/rivaroxaban disposition [167,168]. Consistent with multi-society recommendations on herbal products, discontinuation of curcumin-containing supplements 1–2 weeks before elective procedures is reasonable to minimize additive antithrombotic effects and simplify management [169].

**Resveratrol** demonstrates antiplatelet activity in human/ex vivo models, attenuating thromboxane A_2_–dependent platelet activation and aggregation and reducing platelet activation markers at concentrations achievable with moderate dietary exposure [170,171]. Clinically, bleeding events attributable to resveratrol are sparsely documented, but precaution is warranted: preclinical work shows potentiation of warfarin’s anticoagulant effect, and contemporary clinical reviews flag a plausible bleeding risk—particularly in patients receiving antithrombotic therapy [172,173]. Beyond pharmacodynamics, resveratrol modulates drug disposition pathways—in healthy volunteers, high-dose resveratrol inhibited CYP3A4 activity, and multiple studies indicate interactions with P-glycoprotein—supporting a theoretical interaction with DOACs and other CYP3A4/P-gp substrates even though clinical confirmation is limited [174,175,176]. In perioperative care, expert consensus favors temporary discontinuation 1–2 weeks before elective procedures when used alongside antiplatelet/anticoagulant therapy, given uncertain benefit and potential additive bleeding risk [147,169].

**Omega-3** PUFA consumption decreases arachidonic acid- and thromboxane-derived metabolites from omega-6 pathways, promoting anti-inflammatory and antithrombotic effects. This shift reduces platelet activation and clot formation potential, which may slightly increase bleeding tendency. However, a systematic review and meta-analysis of randomized clinical trials found no association between omega-3 PUFA intake and an increased risk of overall, hemorrhagic, intracranial, or gastrointestinal bleeding. In contrast, high-dose purified EPA was linked to a modest, dose-dependent increase in bleeding risk, independent of antiplatelet therapy [177]. Similarly, another systematic review and meta-analysis reported that EPA monotherapy, compared with control, was associated with a higher risk of total bleeding and atrial fibrillation [178]. The results of clinical studies on adverse events associated with omega-3 fatty acid supplements are summarized in Table 1. Also, a systematic review of randomized controlled trials conducted between 1987 and 2023 found that individuals taking omega-3 polyunsaturated fatty acids (PUFAs) were more likely to experience minor adverse effects, including diarrhea, taste disturbances, and a tendency to bleed, while reporting less back pain. Importantly, no serious adverse events directly linked to omega-3 PUFAs were observed, emphasizing the need for comprehensive monitoring to detect subtle side effects [179].

Another systematic review relevant for surgical practice found no evidence to support discontinuing fish oil supplements prior to surgery or other invasive procedures [180]. Moreover, omega-3 PUFAs have demonstrated benefits in liver surgery and acute respiratory distress syndrome, suggesting potential value in perioperative care for trauma patients by reducing hospital and intensive care unit stays [181].

However, omega-3 use in early pregnancy was associated with a higher risk of postpartum hemorrhage, supporting recommendations to discontinue supplementation in late pregnancy [182]. Despite its benefits, omega-3 supplementation can increase bleeding risk by inhibiting platelet function, particularly in patients on anticoagulants, as illustrated by an elderly patient on warfarin and fish oil whose coagulopathy was uncorrectable after blunt head trauma [183].

**Garlic** extract offers a safe and modestly effective adjunct for lipid management, particularly in individuals with mild hyperlipidemia or those intolerant to statins. While it is not a substitute for statin therapy in high-risk patients, its lipid-lowering, antioxidant, and potential anti-atherogenic effects justify its inclusion in dietary and lifestyle-based approaches to cardiovascular risk reduction [184].

Garlic inhibits platelet aggregation by suppressing cyclooxygenase activity and thromboxane A_2_ synthesis, reducing intraplatelet Ca^2+^, increasing cAMP and cGMP, and decreasing fibrinogen binding and platelet shape change [63]. Garlic compounds may also directly interfere with GPIIb/IIIa receptors, reducing platelet–fibrinogen interactions [60].

Importantly, garlic may exert mild antiplatelet effects, which can increase bleeding risk and may be clinically relevant in patients receiving anticoagulant or antiplatelet therapy. But overall, the use of garlic appeared to be safe among individuals on warfarin [185,186].

**Nattokinase** was shown to be capable of blocking thromboxane formation resulting in an inhibition of platelet aggregation without producing the side effect of bleeding [69].

Thus, nattokinase shows promise for cardiovascular and thrombotic disorders, offering antithrombotic, antihypertensive, antiatherosclerotic, and neuroprotective benefits with a strong safety profile. But, short-term, low-dose nattokinase may not significantly reduce lipid levels, but it shows potential as a supportive therapy for managing hypertension [187].

In a clinical trial involving 113 patients with dyslipidemia, participants were randomly assigned to receive either nattokinase-monascus supplementation or placebo. Treatment with nattokinase-monascus supplementation significantly reduced total cholesterol and low-density lipoprotein cholesterol, while no significant effects on coagulation parameters were observed [71].

Recent studies have demonstrated that NK exerts therapeutic effects against atherosclerotic pathology by modulating the “Lipid and Atherosclerosis” pathway. High-dose NK treatment (900 FU/kg body weight) for six weeks significantly reduced atherosclerotic lesions in ApoE−/− mice. Liver proteomic analysis revealed downregulation of PXDN 9 (peroxidasin) and PNLIP (pancreatic lipase)—proteins associated with lipid oxidation and absorption—leading to enhanced lipid metabolism, reduced lipid peroxidation, and overall attenuation of atherosclerosis. In the same study, the researchers also showed that NK exhibited a dose-dependent thrombolytic effect on both fresh and aged blood clots, achieving a 30–40% thrombolysis rate at a concentration of 5000 μg/mL [188].

Currently, clinical data on the potential risks of long-term nattokinase use—such as bleeding tendency and liver burden—are limited. Although human trials show encouraging results, further research on its pharmacokinetics and drug interactions is needed before it can be considered a substitute for conventional cardiovascular medications. medications [189].

Recent studies suggest that nattokinase, as a functional food ingredient, may benefit aging-related diseases without significant side effects [190].

### 3.2. Micronutrient-Related Coagulation Effects

**Vitamin E** at pharmacologic doses (≥300–400 IU/day) can antagonize vitamin K-dependent coagulation through CYP4F2-mediated pathways and may also inhibit platelet aggregation [191,192]. In anticoagulated patients, higher serum α-tocopherol concentrations have been associated with increased bleeding risk during warfarin or acenocoumarol therapy, and recent case reports describe reversible coagulopathy precipitated by excessive vitamin E intake [193,194]. Consequently, perioperative and anesthesia guidelines recommend avoiding high-dose vitamin E in patients receiving anticoagulants and discontinuing supplementation 1–2 weeks prior to elective procedures [169,195].

Unlike vitamin E, vitamin C has no consistent clinical bleeding signal; a controlled study of high-dose oral vitamin C (500 mg twice daily, 14 days) found no overall significant effect on CYP3A4 activity (a modest induction was seen only in men), and contemporary clinical meta-analyses have not linked vitamin C to excess hemorrhage [196,197].

### 3.3. Pharmacokinetic Interactions with Antithrombotics

**Asian ginseng**, with its diverse ginsenoside profile, poses a potential pharmacokinetic interaction with antithrombotic therapy, primarily through attenuation of warfarin’s anticoagulant effect. In a randomized, double-blind, placebo-controlled trial in healthy volunteers, American ginseng significantly reduced warfarin’s anticoagulant response—reflected by lower INR and diminished pharmacodynamic effect—consistent with increased drug clearance [198]. Clinical probe studies demonstrate that Asian ginseng induces CYP3A activity, reducing oral midazolam exposure with minimal influence on P-glycoprotein (P-gp), a mechanism expected to preferentially decrease R-warfarin exposure and thereby blunt overall anticoagulation [199]. Contemporary reviews corroborate the directionality of this interaction, while noting heterogeneity across species and extract types, and recommend close INR monitoring or avoidance when ginseng is initiated or discontinued in patients receiving vitamin K antagonists [200,201]. For DOACs, direct clinical data are lacking, but potential CYP3A4/P-gp modulation implies a theoretical risk of reduced exposure (notably apixaban/rivaroxaban), warranting caution in high-risk settings and around elective procedures, where many perioperative sources advise discontinuation ~1–2 weeks pre-operatory [200].

**Coenzyme Q10** (ubiquinone) is structurally related to vitamin K, and multiple clinical reports/interaction databases describe reduced warfarin anticoagulation with INR returning to baseline after CoQ10 discontinuation, consistent with a vitamin-K–like antagonism [202]. Mechanistically and clinically, the principal risk is therefore subtherapeutic anticoagulation (↓INR) rather than bleeding, warranting caution when CoQ10 is added to or withdrawn from stable warfarin therapy [186]. Notably, the only randomized, double-blind, placebo-controlled crossover trial in 24 stable VKA-treated outpatients (CoQ10 100 mg/day for 4 weeks) found no significant change in INR or warfarin dose versus placebo—highlighting heterogeneity and dose/preparation effects across the literature [203]. Contemporary references therefore recommend closer INR monitoring or avoidance when initiating or stopping CoQ10 in patients on VKAs [202]. For DOACs, there is no robust clinical data showing a meaningful interaction with CoQ10; given the lack of a clear CYP3A4/P-gp mechanism, any effect is theoretical, so practice should default to clinical vigilance rather than a blanket contraindication [160].

At pharmacologic doses, resveratrol modulates human drug-metabolizing enzymes: in a randomized crossover study of healthy volunteers given 1 g/day for 4 weeks, resveratrol inhibited CYP3A4, CYP2D6 and CYP2C9 activity (phenotypic probe cocktail), a profile that could increase exposure to CYP3A4-metabolized antithrombotics and co-medications [174]. Preclinical work further suggests potentiation of warfarin anticoagulation at higher resveratrol concentrations, underscoring a plausible PK/PD interaction even though definitive clinical bleeding trials are lacking [172]. In parallel, curcumin demonstrably modulates intestinal transport and metabolism: contemporary mechanistic studies show P-glycoprotein inhibition and interaction with CYP3A4, while a human PK trial found oral curcumin reduced bioavailability of a P-gp/CYP3A4 probe substrate via apparent CYP3A4 activation, highlighting formulation- and context-dependent effects [173,204]. Regulatory and clinical safety sources flag turmeric/curcumin as potential contributors to excess anticoagulation/bleeding when combined with warfarin, with case reports of INR elevation and hemorrhage and professional compendia recommending caution [171]. For DOACs, direct clinical data are sparse, but expert reviews note theoretical risk of altered exposure where CYP3A4/P-gp pathways are relevant (e.g., apixaban/rivaroxaban), justifying conservative perioperative practice and careful co-medication review rather than blanket prohibition [205].

**Berberine** is a multifunctional compound with potential benefits in a variety of conditions, including cardiovascular disease, type 2 diabetes mellitus, gastrointestinal disorders, polycystic ovary syndrome, nonalcoholic fatty liver disease, hyperlipidemia, metabolic syndrome, obesity, and schizophrenia. According to the latest evidence—including the first systematic review on berberine published this year—this compound appears to support cardiovascular health by modulating gut microbiota and regulating energy metabolism through the AMPK–PGC1α pathway [206]. Also, recent studies suggest that berberine may help treat atherosclerosis by inhibiting both the proliferation and apoptosis of vascular smooth muscle cells under mechanical stretch, as well as exerting regulatory effects on platelets [207,208].

Paul et al. reported that berberine inhibits platelet aggregation and superoxide production by modulating aldose reductase, NADPH oxidase, and glutathione reductase. Berberine also decreases calcium release, ERK activation, secretion of α- and dense granules, and platelet adhesion. By inhibiting the p38–p53 pathway, it prevents Bax activation, mitochondrial dysfunction, and platelet apoptosis under high-glucose conditions, suggesting potential vascular benefits in diabetes mellitus [207].

Additionally, berberine exhibits anticoagulant and thrombolytic effects by stimulating antithrombin III, lowering plasminogen activator inhibitor-1 (PAI-1), enhancing urokinase and streptokinase activity, and inhibiting platelet aggregation induced by ADP, thrombin, collagen, or arachidonic acid. Experimental studies show that berberine reduces hypercoagulation induced by a high-fat diet and suppresses thrombus formation, underscoring its therapeutic potential against thrombosis in metabolic diseases [209]. In vivo, oral administration of berberine effectively suppressed platelet activation and carrageenan-induced thrombosis in mice, without prolonging bleeding time [48].

In a recent systematic review and meta-analysis of randomized clinical trials, berberine monotherapy lowers blood lipids and glucose, improves insulin resistance, and shows stronger effects after more than three months of treatment, proving beneficial for older adults with metabolic disorders. Berberine is considered clinically safe and well-tolerated in humans, with few reported adverse effects and no observed impact on participants’ dietary habits [210].

In an umbrella review on berberine and health outcomes, potential side effects identified include constipation, diarrhea, abdominal bloating, and a bitter taste [211].

Several clinical trials (NCT01697735) on berberine 500 mg, administered twice daily as monotherapy or in combination with statins, are ongoing or recently completed. However, the number of participants remains small, and the results have not yet been made publicly [212].

Berberine hydrochloride (produced by the National Institute for the Control of Pharmaceutical and Biological Products) competes to bind warfarin-binding plasma proteins. So, Berberine can displace warfarin from plasma proteins, increasing its free concentration and bleeding risk [213].

**Red Yeast Rice (RYR)** has been shown in studies to potentially improve outcomes after major surgeries, being associated with fewer postoperative complications, lower mortality rates, and reduced hospital stay and costs [214]. However, a single case reported by Mazzanti et al. indicated a possible interaction between RYR supplementation and an anticoagulant, although treatment was not discontinued in that instance to confirm causality [215].

RYR extracts with hepatotoxic properties may form during maturation. Additionally, if poorly manufactured, red yeast rice can contain citrinin, a mycotoxin with hepatotoxic and nephrotoxic potential [216].

Because monacolin K acts like a statin, it carries risks of drug–drug interactions, especially with CYP3A4 inhibitors, leading to possible serious side effects [217].

Although RYR extract is generally considered safe and has not been linked to life-threatening or frequent adverse events or an increased risk of musculoskeletal disorders its safety remains debated [52,218]. The EFSA Panel on Nutrition, Novel Foods and Food Allergens (NDA) recently reassessed the safety of monacolins from RYR under EU scrutiny (Regulation (EC) No 1925/2006, Part C, Annex III). Based on updated analytical, toxicological, and clinical evidence, the Panel reaffirmed earlier concerns (EFSA ANS Panel, 2018) that exposure to monacolin K—even at doses as low as 3 mg/day—may lead to serious hepatic and musculoskeletal adverse effects, including rhabdomyolysis. Therefore, the safety of RYR supplements remains inconclusive and continues to be the subject of scientific debate [219].

### 3.4. Agents with Limited or No Consistent Clinical Bleeding Signal

**Astaxanthin** has not been associated with a consistent clinical bleeding signal in human trials, and regulatory safety evaluations deem typical intakes well tolerated; nonetheless, caution is advisable in patients receiving anticoagulant or antiplatelet therapy [220].

Table 2 summarizes the multi-mechanistic actions of selected anti-atherosclerotic nutraceuticals—highlighting hypolipidemic, vasodilatory, and anti-inflammatory pathways alongside fibrinolytic, antiplatelet, and anticoagulant effects.

### 3.5. Extracellular Vesicles at the Thrombosis–Bleeding Interface

In atherosclerosis, extracellular vesicles (EVs) released by endothelial cells, platelets, smooth-muscle cells, and macrophages act as key mediators of intercellular communication. By exposing phosphatidylserine and tissue factor, and carrying adhesion molecules, lipids, and regulatory RNAs, they amplify vascular inflammation, leukocyte recruitment, and thrombin generation, contributing to plaque progression and rupture. Conversely, distinct EV profiles may also reflect plaque stabilization and are being explored as biomarkers and therapeutic carriers. Anti-atherosclerotic nutraceuticals such as omega-3 fatty acids, curcumin, resveratrol, ginkgo, garlic, berberine, and nattokinase may influence EV biogenesis or cargo, potentially modulating platelet reactivity and bleeding risk when combined with antithrombotic drugs. Including EV-related endpoints in nutraceutical safety studies could help clarify these mechanisms and refine risk assessment [221,222].

## 4. Future Research Directions

Future research on nutraceuticals should prioritize methodological rigor, standardization, and reproducibility. Well-designed randomized controlled trials with sufficient sample sizes, clearly defined populations, and standardized formulations are essential to confirm efficacy and safety. Attention should be given to dose–response relationships, bioavailability, treatment duration, and interactions with conventional therapies. Long-term safety and possible rebound effects after discontinuation warrant further investigation.

Ensuring product quality and consistency is crucial, as variability among commercial preparations often limits comparability and reliability of findings. Mechanistic studies are also needed to clarify the molecular pathways underlying the metabolic, anti-inflammatory, and vascular effects of nutraceuticals.

Collaborative efforts among clinicians, nutrition scientists, and regulatory agencies will be key to establishing evidence-based guidelines. Overall, future studies should provide robust evidence to support the safe and effective integration of nutraceuticals into clinical practice as complementary tools for improving cardiometabolic and overall health outcomes.

## 5. Conclusions

Omega-3 fatty acids and garlic demonstrate modest lipid-lowering effects, with purified EPA showing more consistent cardiovascular benefits than mixed EPA and DHA formulations. Nattokinase and red yeast rice (RYR) may also support cardiovascular health, though their efficacy depends heavily on formulation, dosage, and manufacturing quality.

Vitamin C, vitamin E, resveratrol, astaxanthin, and coenzyme Q10 may offer vascular benefits but can interfere with coagulation-related therapies through effects on hemostasis, drug metabolism, or redox-sensitive pathways. Their use alongside antithrombotic agents or CYP3A4-metabolized drugs should therefore be individualized and undertaken with caution.

Ginkgo biloba, ginger, ginseng, and curcumin show promising anti-atherosclerotic and vascular anti-inflammatory properties, but all carry a clinically significant risk of increased bleeding, particularly when combined with antiplatelet or anticoagulant therapy. While mechanistic data supports their vascular benefits, inconsistent evidence, variability in formulations, and the lack of standardized clinical trials highlight the need for caution and further research before routine clinical use.

Strict adherence to Good Manufacturing Practices is essential to ensure the safety and reproducibility of supplements.

## Figures and Tables

**Figure 1 ijms-26-10183-f001:**
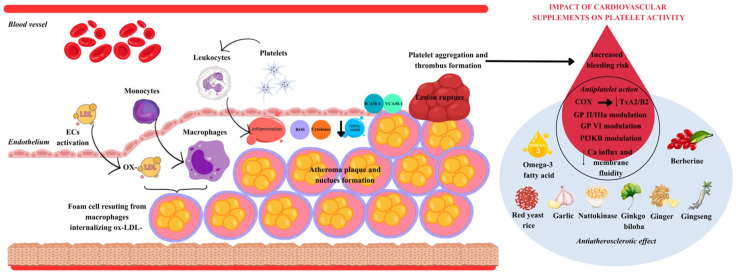
The overlap between anti-atherosclerotic mechanisms and antiplatelet effects.

**Table 1 ijms-26-10183-t001:** Clinical Trials Evaluating the Cardioprotective Role of Omega-3 Fatty Acid Supplements.

Feature	ASCEND Trial [39]	REDUCE-IT Trial [40,41]	STRENGTH Trial [42]	VITAL Trial [43]
Patient Characteristics	Diabetic, no prior CV events	Diabetic with additional CV risk factors or patients with established CVD	High-risk CVD patients, statin-treated	Generally healthy, men ≥ 50 yrs, women ≥ 55 yrs
Number of Patients	15,480	8179	13,078	25,871
Dosage	1 g/day Omega-3 (EPA 460 mg + DHA 380 mg) and placebo -a subgroup with 100 mg/day Aspirin	4 g/day Icosapent Ethyl (EPA only) and placebo-mineral oil	4 g/day Omega-3 (EPA + DHA) and placebo—corn oil	1 g/day Omega-3 (Omacor, fish oil) with or without Vitamin D3: 2000 IU/day and placebo
Follow-up Duration	~7.4 years	4.9 years	~3.5 years	5.3 years
Main Findings	Aspirin reduced CV events Omega-3 showed no CV benefit	Strong evidence supporting this dose of EPA for reducing cardio-vascular events in high-risk patients	Omega-3 showed no CV benefit	Omega-3 showed no reduction in major cardiovascular events, but a modest reduction in myocardial infarction in those with low baseline fish intake
Mechanism of action for cardiovascular benefits	Aspirin: antithrombotic Omega-3: the relatively low dose may have been inadequate to produce meaningful cardiovascular benefit	Reduced triglycerides anti-inflammatory improved endothelial function plaque stabilization antioxidant antithrombotic	DHA may counteract EPA’s beneficial effects, so formulation rather than dose explains the lack of cardiovascular benefits	Omega-3: the relatively low dose may have been inadequate to produce meaningful cardiovascular benefit
Safety/ Adverse Effects	Increased major bleeding after aspirin intake No major issues with omega-3	Increased atrial fibrillation risk, a trend toward higher serious bleeding risk	Increased atrial fibrillation risk, gastrointestinal side effects	No major safety concerns

**Table 2 ijms-26-10183-t002:** Anti-atherosclerotic mechanisms of selected nutraceuticals; hemostatic actions (antiplatelet, fibrinolytic) highlighted in red.

Compound	Key Mechanisms/Actions	References
Omega-3 fatty acids (EPA, DHA)	↓ Triglycerides, modest ↑ HDL-C	[27]
	Hepatic de novo lipogenesis and increased postprandial fatty acid oxidation	[39]
	↑ Resolvin = anti-inflammatory effects	[32]
	↓ TxA_z_ synthesis & GPVI modulation, ↓ Ca^2+^ influx & membrane fluidity	[29,32,36]
Berberine	↓ Triglycerides and LDL-C, modest ↑ HDL-C	[46]
	↓ ROS anti-inflammatory effects	[47]
	Improves insulin sensitivity	[45]
	Blocks GPIIb/IIIa, modulates PI3Kβ/Ca^2+^	[48]
Red yeast rice	Monacolin K acts like a statin = ↓LDL-C	[50]
	↓ TXB2 and ↑ antithrombin III	[51]
Garlic	↓ TC, ↓ TG, ↓ LDL-C slightly ↑ HDL-C	[57]
	↓ ROS, ↓ NF-kB, ↑ NO, ↑ H_2_S ↑ ANP, ↓ SRAA, ↑ VSMC proliferation ⟶ vasodilation and lower blood pressure	[62]
	Decrease the absorption of cholesterol, -HMG-CoA Reductase	[53]
	Increased secondary bile acids, increase GLP-1	[61]
	Inhibits platelet activation & GP IIb/IIIa binding	[63]
	Inhibits fibrinogen binding and platelet shape change	[60]
Nattokinase (NK)	↓ TG, ↓ LDL-C, ↓ ox LDL, ↑ HDL-C	[188]
	Down-regulated PXDN and PNLIP	[189]
	-HMGcoA reductase, +LPLase	[64]
	↓ ROS, ↓ IL-6	[65]
	↓ Blood pressure	[71]
	↑ tPA release, ↓ TXA_2,_ ↓ Fibrinogen & clotting factors (VII, VIII)	[67,69,70]
Ginkgo biloba	↑ NO bioavailability	[114]
	↓ ICAM-1, VCAM-1	[115]
	↓ ROS, ↓cytokines	[116]
	↓ PAF	[150,151]
Ginger	↓ CRP, ↓ Il-6, ↓ TNFα	[126]
	↓ ROS	[124]
	↓ TXA_2_ GPVI modulation	[157,158]
Ginseng	↑ NO	[131]
	↓ ICAM-1, VCAM-1	[132]
	↓ GPIIb/IIIa, ↓ TXB_2_, GPVI modulation	[133]

Legend: ↑ = increase and ↓ = decrease. The abbreviations are defined at the end of the article.

## Data Availability

No new data were created or analyzed in this study. Data sharing is not applicable to this article.

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
