# Peer review of "Navigating the Effects of Anti-Atherosclerotic Supplements and Acknowledging Associated Bleeding Risks"

_ijms, 2025, doi:10.3390/ijms262010183_

Round 1
Reviewer 1 Report
Comments and Suggestions for Authors
This article (ijms-3891655) sought to review the major anti-atherosclerotic supplements and their associated bleeding risks. Five lipid-lowering supplements (i.e., omega-3 fatty acids, berberine, red yeast rice, garlic, and nattokinase), 5 LDL oxidation-inhibiting supplements (i.e., astaxanthin, resveratrol, coenzyme Q10, vitamin C, and vitamin E), and 4 anti-inflammatory supplements (i.e., ginkgo biloba, ginger, ginseng, and curcumin) were discussed in this article. It seems that this manuscript must be improved greatly.
Major comments:
- Honestly speaking, I do not like the present Introduction section (particularly the parts after the third paragraph, i.e., lines 55-116) which describes too many details (or background) not closely related with the main topic.
- It would be better to display the main mechanisms of atherogenesis and bleeding risks via a schematic diagram and describe the main mechanisms in the Introduction section.
- It would be better to separate section 2 into two sections, e.g., section 2 (anti-atherosclerotic potential) and section 3 (associated bleeding risks) because they are the two focuses of this review according to the manuscript title. In the present manuscript, the bleeding risk topic is out of focus (some subsections even do not talk about bleeding risks).
- There are too many paragraphs in some subsections making the text unfriendly to readers (it seems that all information is piled together disorderly). It would be better to reorganize each subsection more logically making the text more readable.
- It would be better to summarize the clinical trials (e.g., subsection 2.1.1.) via a table
- A section of perspectives or research trends should be added prior to the Conclusion section.
Minor comments:
- When at first mention, an abbreviation instead of its full name should be put in parentheses, for example, “LDL-C (low density lipoprotein - cholesterol” (line 44 on page 2) should be changed to “low density lipoprotein – cholesterol (LDL-C)”. Please check throughout the main text, e.g., ADAMTS13 (line 73), Mac-1 (line 87), LFA-1 (line 88), PF4 (line 89), VEGF (line 95), VCAM and ICAM-1 (line 99), etc.
- The two words platelet and thrombocyte appear alternately. Please use only one of them (just mention the other one at first mention) throughout the main text.
- Representative references should be provided at the corresponding locations in all tables supporting the effects/mechanisms described. It may also help readers find the references easily and rapidly.
Reviewer 2 Report
Comments and Suggestions for Authors
This article reviews the potential roles of various dietary supplements and herbal medicines, including omega-3 fatty acids, garlic, ginkgo biloba, ginseng, ginger, red yeast rice, and vitamins, in the prevention and management of atherosclerosis and cardiovascular diseases. It discusses their mechanisms of action at molecular, endothelial, and platelet levels, evaluates existing evidence on their efficacy, and addresses safety considerations, including drug interactions and bleeding risks. The review highlights the importance of high-quality clinical research and standardized manufacturing practices in ensuring the safety and reproducibility of these supplements in clinical settings. According to this article, there are several weaknesses that need to be addressed.
Specific comments
- The article suggests that several botanical and nutritional supplements have potential anti-atherosclerotic effects, but evidence for their clinical effectiveness is limited. It recommends supplementing this with data from high-quality clinical trials to strengthen the credibility of these findings.
- Many supplements, such as Ginkgo biloba, Ginseng, and Curcumin, have potential benefits, but their bleeding risks and drug interactions are not discussed clearly.
- Most of the literature comes from a small number of studies or animal experiments, lacking large-scale, multicenter randomized controlled trials to support its clinical application. It is recommended to increase evidence from multiple sources to improve the comprehensiveness of the topic.
- Incorporating an abstract illustration that conveys the central theme of the article would help engage readers more effectively and strengthen their understanding of the key points.
Minor comments
- The article format needs to be checked.
- In line 653, Spelling errors, for example, "Henorrhagic" should be "Hemorrhagic".
- Plant names should be in italics or regular script, and the format should be consistent.
Round 2
Reviewer 1 Report
Comments and Suggestions for Authors
The revised manuscript has addressed most of my previous comments. I still have the following minor comments.
1. Is it possible to incorporate some major or important supplements into Figure 1 added in the revised manuscript?
2. It has been widely reported that extracellular vesicles derived from vascular cells (including platelets and endothelial cells) and diseases play important role in atherosclerosis (please refer to “Y. Zhang, et al. Diversity of extracellular vesicle sources in atherosclerosis: role and therapeutic application. Angiogenesis. 2025. 28: 34” and “F. Fang, et al. Extracellular vesicles in atherosclerosis: From pathogenesis to theranostic applications. Small. 2025. 21: 2504761”). It is possible that extracellular vesicles are involved in the effects of anti-atherosclerotic supplements and/or associated bleeding risks. It would be better to briefly introduce the involvement of extracellular vesicles. Linking with extracellular vesicles will help increase the citation of this article because extracellular vesicles presently are one of the hottest research topics in biology.
